biochemistry/environmental science/health and disease and epidemiology

*Aedes aegypti*, insecticide resistance, temephos, Bti, growth inhibitors, Jamaica

**Authors for correspondence:**
Sheena Francis
e-mail: sheena.francis02@uwimona.edu.jm
Carolina Torres Gutierrez
e-mail: aniloract@gmail.com

# Comparative toxicity of larvicides and growth inhibitors on *Aedes aegypti* from select areas in Jamaica

Sheena Francis[1,2,3], Jervis Crawford[3], Sashell McKenzie[3], Towanna Campbell[3], Danisha Wright[1,3], Trevann Hamilton[3], Sherine Huntley-Jones[4], Simone Spence[5], Allison Belemvire[6], Kristen Alavi[7] and Carolina Torres Gutierrez[8]

[1]Natural Products Institute, University of the West Indies, Mona, Jamaica
[2]Abt Associates, 70 Half-Way Tree Road, Kingston, Jamaica
[3]Zika AIRS Project Jamaica, 70 Half-Way Tree Road, Kingston, Jamaica
[4]Vector Control Unit, Ministry of Health and Wellness, Kingston, Jamaica
[5]Health Promotions and Protection, Ministry of Health and Wellness, Kingston, Jamaica
[6]United States Agency for International Development (USAID), Bureau for Global Health, Office of Infectious Disease, Malaria Division, Arlington, TX, USA
[7]United States Agency for International Development (USAID), Washington, DC, USA
[8]International Development Division (IDD), Abt Associates, Rockville, MD 20852, USA

SF, 0000-0002-2240-8627; CTG, 0000-0002-3488-1745

Insecticide resistance has become problematic in tropical and subtropical regions, where *Aedes* mosquitoes and *Aedes*-borne arboviral diseases thrive. With the recent occurrence of chikungunya and the Zika virus in Jamaica, the Ministry of Health and Wellness, Jamaica, partnered with the United States Agency for International Development to implement multiple intervention activities to reduce the *Aedes aegypti* populations in seven parishes across the island and to assess the susceptibility of collected samples to various concentrations of temephos, *Bacillus thuringiensis* subsp. *israelensis,* (Bti), diflubenzuron and methoprene. Of the insecticides tested, only temephos has been used in routine larviciding activities in the island. The results showed that only temephos at concentrations 0.625 ppm and Bti at concentrations 6–8 ppm were effective at causing 98–100% mortality of local *Ae. aegypti* at 24 h exposure. Surprisingly, the growth inhibitors diflubenzuron and methoprene had minimal effect at preventing adult emergence in *Ae. aegypti* larvae in the populations tested. The results demonstrate the need for insecticide resistance testing as a routine part of vector control

monitoring activies in order to determine useful tools that may be incorporated to reduce the abundance of *Ae. aegypti*.

# 1. Introduction

The *Aedes aegypti* mosquito is the most widespread vector found throughout the Caribbean [1] and transmits several illnesses such as dengue and urban yellow fever. In recent years, Jamaica has had several occurrences of vector-borne diseases, from the endemic dengue virus [2] to novel diseases such as chikungunya and Zika [3,4]—diseases caused by viral agents and transmitted by *Aedes* mosquitoes. In an effort to decrease the burden of Zika transmission in the country, the Ministry of Health and Wellness (MOHW) of Jamaica partnered with the United States Agency for International Development (USAID)-funded Zika AIRS Project (ZAP). ZAP sought to build the capacity of stakeholders at both the ministerial and national level on vector control strategies, identified and treated potential breeding sites and conducted entomological monitoring, including insecticide susceptibility tests. ZAP was primarily situated in the seven most eastern parishes in the island, engaging 94 communities and routinely inspecting and manually applying a biological larvicide to 36 000 premises on a monthly basis.

Chemical control is the main method of vector management in most tropical countries; however, its effectiveness is limited by the method of application employed to cover large areas [5] and the development of resistance to insecticides in the local vector populations. Resistance to insecticides in mosquitoes is one of the main hindrances to the control of hematophagous insects that are of concern to public health [6], and it is an issue that has already been reported in Jamaica [7] and observed in the region [8,9]. Given the significant impact of insecticide resistance on vector management, routine assessment of chemical resistance and the incorporation and rotation of insecticides with varied modes of mechanism [10,11] are important activities that can be incorporated in integrated vector management. These practices within Jamaica would ensure that any vector control strategy to be used by the national authorities is tailored to guide subnational strategies to suit local context in countries like Jamaica.

The present study evaluated the susceptibility of wild populations of the mosquito vector *Ae. aegypti* to temephos, a larvicide extensively used in Jamaica [1], as well as a biolarvicide based on *Bacillus thuringiensis* subsp. *israelensis* (Bti), and insect growth regulators (IGR) such as methoprene and diflubenzuron—these latter products with no known history of use on the island. We present a wide-scale evaluation of the current susceptibility status of the mosquito vector *Ae. aegypti* to larvicide products in the eastern parishes of Jamaica.

# 2. Material and methods

## 2.1. Study site

The study was carried out in communities or neighbourhoods of seven parishes of Jamaica, located in the southeastern (St Catherine, Kingston and St Andrew, St Thomas) and northeastern (St Mary, St Ann and Portland) regions of the island. Jamaica is the third largest island in the Caribbean, with a population of 2.93 million people [12]. Jamaica is a tropical island with average temperatures fairly constant throughout the year, oscillating from 25 to 30°C in the lowlands and 15 to 22°C at higher elevations. The rainy season occurs from late April to October. ZAP project conducted activities focused on two main goals: entomological surveillance and vector control. The insecticide susceptibility testing of *Ae. aegypti* populations was included in the entomological surveillance component of the implementation. The project conducted operations in Jamaica during two phases: from April 2018 to July 2018 (Phase I) and from September 2018 to April 2019 (Phase II). Mosquito sampling occurred during active operations in the field (Phase II). All field operations were agreed and in close collaboration with the Ministry of Health and Wellness of Jamaica.

## 2.2. Materials

With the exception of the biolarvicide, *Bacillus thuringiensis* subsp. *israelensis*, Strain AM65-52 (Vectobac WDG®) purchased from Valent BioSciences (IL, USA), all larvicides used in this study were complete

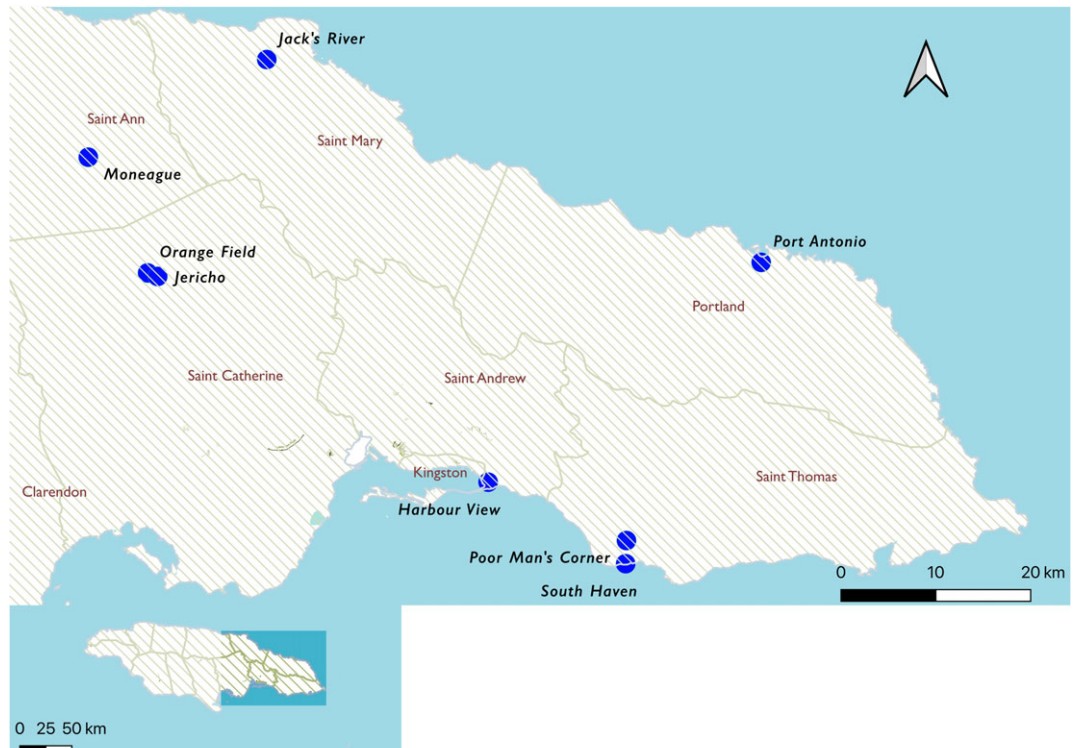

**Figure 1.** Locations of sentinel sites established in this study (created using QGIS v. 3.10).

kits. Temephos, diflubenzuron and methoprene, along with the ethanol control, were procured from the Universiti Sains Malaysia (USM), Vector Control Research Unit, Infotech (Pinang, Malaysia), the only agency approved by the World Health Organization (WHO) to supply insecticide kits and materials for regular surveillance in public health.

A reference strain of *Ae. aegypti*, designated as Rockefeller strain, was used as the known susceptible population to all bioassays. This strain was generously donated by the Centers for Disease Control and Prevention (CDC, Atlanta, GA, USA). The biological material was donated as eggs that were reared under insectary conditions and maintained isolated during the entire testing period.

### 2.2.1. Oviposition

The larvicide susceptibility of wild *Ae. aegypti* populations was analysed, including samples from seven of the eastern parishes in Jamaica, namely, St Catherine, Kingston and St Andrew (KSA), St Thomas, St Mary, St Ann and Portland. Sentinel sites were established in 100 homes in each of the parishes (figure 1) through the period of 17 September–7 December 2018 (permission to establish sentinel sites in homes was acquired prior to the initiation of work). Six wild populations of *Ae. aegypti* were established under insectary conditions, with one population per parish, with Kingston and St Andrew represented as one single population of *Ae. aegytpi*, given their spatial proximity.

Black oviposition cups (450 ml capacity × 14.5 cm tall) were lined with pellon paper (7.5 × 30.5 cm) and filled with 50 ml water containing 5 g of instant yeast (Lesaffre, France). The ovitraps were checked and replaced weekly. Papers from the traps were collected weekly and carried to the Mosquito Control Research Unit (MCRU), located in Mona campus of the University of the West Indies in Kingston. Once in the laboratory, the egg papers were allowed to air dry (for around 2–3 days) and assessed for the presence of *Aedes* eggs which were then counted for surveillance and record keeping purposes. Egg papers were periodically set to hatch for bioassays under insectary conditions.

### 2.2.2. Mosquito rearing

The egg papers (25 papers per parish) were submerged in acrylic containers (37.5 × 25 × 15 cm) containing 3 l of water with 0.1 g of instant yeast (Lesaffre, France). The activation of eggs followed the methodology described in Zheng *et al.* [13] to increase hatching. Briefly, tap water was collected,

allowed to stand for a minimum of 48 h, boiled, covered and allowed to cool at room temperature for 24 h prior to adding the yeast and egg papers. The containers with the egg papers were then covered until larvae emergence was observed. The larvae were maintained under standard rearing conditions: $25 \pm 2^{\circ}C$, $75 \pm 5\%$ relative humidity and 12 : 12 h light : dark photoperiod. The larvae were fed on a diet of ground cat food (Friskies®) and observed daily to their third instar stage.

The larvae from Portland were field caught and donated by the Ministry of Health and Wellness of Jamaica to the project. The larvae were reared as described above. The insectary facilities employed by this study correspond to the recently inaugurated Mosquito Control and Research Unit at the University of West Indies, Mona campus, in Kingston, Jamaica.

## 2.3. Bioassays

The late third instar *Ae. aegypti* larvae populations from each parish were tested for their susceptibility to the organophosphate temephos, the biolarvicide *Bacillus thuringiensis* subsp. *israelensis* (Bti) or the IGR diflubenzuron or methoprene. All assays were conducted as stipulated by the WHO guidelines [14]. These assays were conducted on *Ae. aegypti* populations collected from St Catherine, Kingston and St Andrew, St Thomas, St Mary, St Ann and Portland, Jamaica. All bioassays were conducted in quadruplets (four replicates), with the exception of Portland. Portland assays were conducted as triplicates owing to limitations in sample availability.

## 2.4. Temephos assays

Stock concentrations of temephos 1.25, 6.25, 31.25 and 156.25 ppm (mg l$^{-1}$) were diluted with distilled water to reach final concentrations of 0.005, 0.025, 0.125 and 0.625 ppm in disposable paper cups containing 20–34 larvae. The 0 ppm containing 0.4% ethanol (ethanol from the kit control) was used as the control. Each assay was observed up to 1 h, and then further at 24 h. Larvae were not fed during the period of observation [14].

## 2.5. *Bacillus thuringiensis israelensis* assays

An amount of 3000 mg of Bti (VectoBac WDG) was mixed in 300 ml of distilled water to prepare a 10 000 ppm (mg litre$^{-1}$) stock solution of Bti. The stock solution was vigorously stirred using a magnetic stirrer (C MAG HS 7; IKA) and aliquoted to give concentrations of 0, 2, 4, 6 and 8 ppm of Bti in 2000 ml of distilled water. Two hundred and fifty millilitres of each concentration were poured into disposable paper cups containing 20–34 larvae; care was taken to agitate the mixture prior to each pour to ensure particulates remained in suspension. The larvae were observed initially at 1 h and then again at 24 h without food [14].

## 2.6. Growth inhibitor susceptibility, diflubenzuron and methoprene assays

Stock concentrations of the IGR, diflubenzuron or methoprene, of 0.0325, 0.16, 0.8, 0.4 and 20 ppm (mg l$^{-1}$) were diluted with distilled water to 0.00013, 0.0064, 0.0032, 0.016 and 0.08 ppm in disposable paper cups containing 20–25 larvae. The 0 ppm containing 0.4% ethanol (ethanol kit control) was used as the control. Larvae were periodically fed through the experiment and observed until total emergence and/or death in the control. The results of each replicate were pooled per concentration to obtain total emergence per concentration. Total emergence per treatment concentration for each population was tallied. The data are presented as total per cent inhibition of emergence per population ($IE\% = C - (E/100) \times 100$), where $C$ is equal to the total per cent emergence in the control group and $E$ is equal to the total per cent emerged in the treatment group. Where $C$ is greater than 5%, but less than 20%, Abbott's formula was used to correct mortality [14].

## 2.7. Data analysis

The data are presented as mean ± s.e. per population for the insecticide temephos and the biolarvicide Bti. Abbott's formula [14–16] was used to correct the mortality rate in each treated group when necessary.

% Corrected Mortality $= ((T - C)/(100 - C)) \times 100$; whereby $T$ is equal to the total per cent mortality in the treated group, and $C$ is equal to the per cent mortality in the control group, providing that the control mortality was less than or equal to 20%.

**Table 1.** Mortality of *Ae. aegypti* larvae exposed to temephos 0–0.625 ppm after 1 h exposure. Mortality of the late third instar *Ae. aegypti* larvae ($N = 60$–100) from the eastern parishes of Jamaica (St Catherine, KSA, St Thomas, St Mary, St Ann and Portland) exposed to varying concentrations (0–0.625 ppm) of temephos at 1 h in comparison to the susceptible Rockefeller strain. The values are expressed as mean ± s.e. Tukey *post hoc* test was used to analyse the significant differences between Jamaican populations at concentrations 0.125 and 0.625 ppm. Means with different letters in the same column are significantly different at $p < 0.01$.

| populations | [temephos] ppm | | | | |
| --- | --- | --- | --- | --- | --- |
| | 0.00 | 0.005 | 0.025 | 0.125 | 0.625 |
| Rockefeller | 0.00 ± 0.00 | 61.66 ± 2.76 | 97.50 ± 2.17 | 100.00 ± 0.00 | 100.00 ± 0.00 |
| St Catherine | 0.00 ± 0.00 | 0.00 ± 0.00 | 0.00 ± 0.00 | 0.00 ± 0.00 a | 0.00 ± 0.00 a |
| KSA | 0.00 ± 0.00 | 0.00 ± 0.00 | 0.00 ± 1.00 | 7.22 ± 1.91 a | 3.70 ± 5.42 a |
| St Thomas | 0.00 ± 0.00 | 0.01 ± 0.01 | 0.00 ± 0.00 | 18.00 ± 2.58 b | 40.00 ± 9.31 b |
| St Mary | 0.00 ± 0.00 | 0.00 ± 1.00 | 0.00 ± 1.15 | 0.00 ± 2.52 a | 29.33 ± 7.30 b |
| St Ann | 0.00 ± 0.00 | 0.00 ± 0.00 | 0.00 ± 0.00 | 1.00 ± 1.00 a | 5.00 ± 2.52 a |
| Portland | 0.00 ± 0.00 | 0.00 ± 0.00 | 0.00 ± 0.00 | 3.33 ± 2.72 a | 6.66 ± 1.36 a |

The mortality in the control group for either the temephos or Bti bioassays per population was less than 20%. Less than 5% pupae were observed after 24 h in the control or exposure groups.

As previously mentioned, for the IGR, data are presented as total mortality per concentration for either diflubenzuron or methoprene.

All statistical analyses were completed using SPSS for Windows (v. 17.0). One-way analysis of variance and *post hoc* Tukey test were used to analyse significant differences ($p < 0.05$) between tested populations in susceptibility to the insecticide.

# 3. Results

## 3.1. Temephos susceptibility

The mortality at 1 h for the Rockefeller strain was markedly different to the tested Jamaican populations. Though there was a positive correlation between time of exposure and increase in concentration of temephos with the larvae from the eastern parishes of Jamaica, minimal toxicity was observed at 1 h exposure to temephos 0.005–0.625 ppm. The mean mortality within 1 h exposure at the highest concentration (0.625 ppm) of temephos for the Jamaican population varied between 0 and 40%. Significant differences in efficacy were $p < 0.001$ (table 1).

In comparison to the laboratory susceptible Rockefeller strain, where $97.5 ± 2.17\%$ died at concentration 0.025 ppm, only 26–78% of the Jamaican populations tested died within 24 h at that concentration. The mortality was greater than 88% in larvae exposed to temephos for 24 h at concentrations 0.125 and 0.625 ppm (figure 2). However, temephos at 0.625 ppm had significantly less ($p < 0.005$) effect on the St Ann population than populations from the other eastern parishes.

## 3.2. *Bacillus thuringiensis israelensis* (Bti) susceptibility

The exposure to the biolarvicide Bti at 6 and 8 ppm resulted in greater than 98% mortality for both the laboratory reference strain (Rockefeller) and the Jamaican test populations within 1 h exposure (figure 3). No significant differences in mortality were observed at these concentrations.

## 3.3. The growth regulator diflubenzuron and methoprene susceptibility

Diflubenzuron varyingly prevented the moulting or transitioning of *Ae. aegypti* larvae to its pupae or adult form resulting in death of the organism. Rarely, an organism displaying the intermediate transition from larvae to pupae was observed at 0.08 ppm diflubenzuron after 48 h exposure (figure 4*a*,*c*: specimen from the St Ann and St Catherine population, respectively) or 10 days (figure 4*b*: specimen from the St Mary population). In figure 4*a*, the pupae were restricted by the incomplete moulting of its larval exoskeleton. In figure 4*b*,*c*, moulting appeared to be restricted to the anterior portion of the organism,

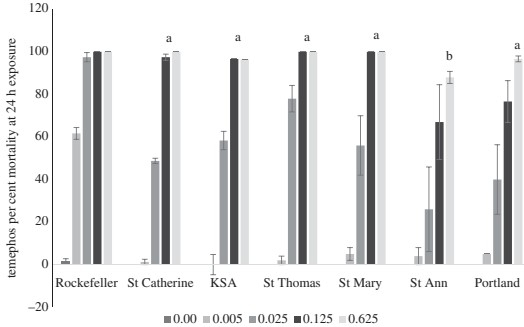

**Figure 2.** *Aedes aegypti* larvae mortality at 24 h exposure to varying concentrations of temephos. The mortality of the late third instar *Ae. aegypti* larvae (*N* = 60–107) from the eastern parishes (St Catherine, KSA, St Thomas, St Mary, St Ann and Portland) exposed to varying concentrations (0–0.625 ppm) of temephos and observed for 24 h in comparison to the susceptible Rockefeller strain. The observed mortality at 0.625 ppm was compared across the test populations. The values are expressed as mean ± s.e. The Tukey *post hoc* test was used to identify differences in the mean at 0.625 ppm temephos for the Jamaican populations. Means at 0.625 ppm temephos with different letters are significantly different at *p* < 0.005.

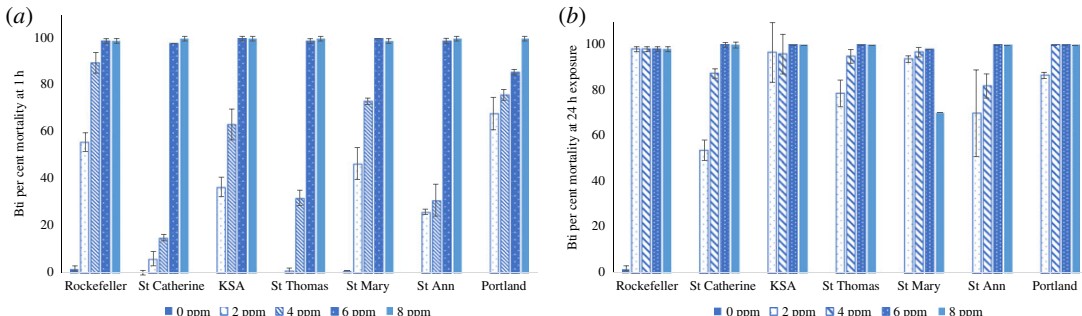

**Figure 3.** *Aedes aegypti* larvae exposure to varying concentrations of *Bacillus thuringiensis israelensis* (Bti). The impact of varying concentrations (0–8 ppm) of Bti for populations from the eastern parishes (St Catherine, KSA, St Thomas, St Mary, St Ann and Portland) in comparison to the susceptible Rockefeller strain after 1 h (*a*) or 24 h (*b*) on *Ae. aegypti* larvae, $F_0$ generation (*N* = 70–120). The data are summarized as mean ± s.e.

where the distinctive pupae 'trumpets' (blue arrow) are seen; however, the rest of the organism is that of a larvae, with its siphon and anal gills at the caudal end (red arrow) of the organism. In comparison, figure 4*d* shows a normally moulted pupae exposed to 0.00 ppm diflubenzuron. All partially moulted pupae died within 24 h of their partial transition.

In figure 5*c*, the legs of the emerging adult remained fused to its pupae casing, causing the winded form to struggle to free itself, which eventually resulted in death. These types of incomplete transitions from pupae to adult form were observed in all populations exposed to 0.08 ppm diflubenzuron.

Both IGR at concentrations of 0.00013–0.16 ppm proved ineffective against the Jamaican test populations. However, concentrations as low as 0.0032 ppm for either IGR resulted in mortality greater than or equal to 96% in the Rockefeller strain. For the Jamaican populations, there was no discernable trend between inhibition of emergence and IGR concentrations. Diflubenzuron at 0.08 ppm resulted in greater than 89% inhibition of emergence in all test populations (table 2). Larvae exposed to this concentration remained in a prolonged larval state until disintegrated. However, exposure to methoprene at 0.08 ppm prevented 16–100% emergence in the test populations (table 3).

# 4. Discussion

Routine entomological surveillance should be a consistent goal for vector control operations of any national public health programme. A complete approach for entomological surveillance must include regular testing of the susceptibility of vector species to the products at hand such as insecticides, larvicides or other alternative tools (e.g. IGR) for the reduction in medically important insect populations. Current guidelines announced by the international authorities providing technical support for the control of *Aedes*-borne diseases around the globe [17] have clearly stated that insecticide susceptibility profiles

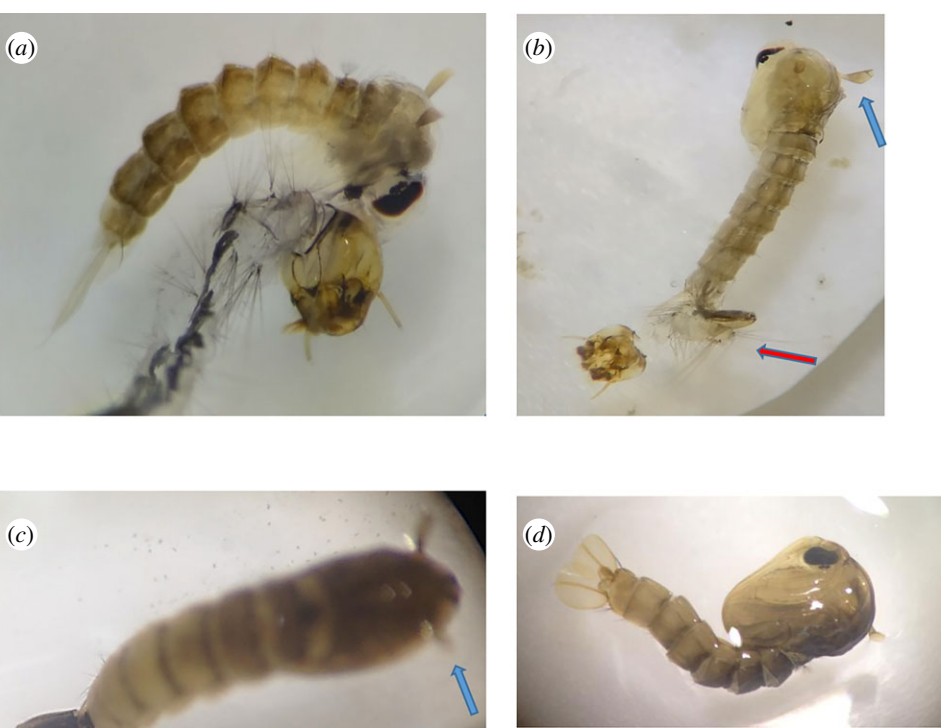

**Figure 4.** *Aedes aegypti* larvae exposed to IGR diflubenzuron at 0.08 ppm. (*a–c*) The partial transition from larvae to pupae in *Ae. aegypti* exposed to 0.08 ppm diflubenzuron. The blue arrows show the presence of the respiratory tubes or trumpets on the cephalothorax of the pupae. The red arrows show the retention of the larval siphon on the caudal end of the pupae. These photos were taken from populations tested during this study. (*d*) A normal pupae formation in 0.00 ppm diflubenzuron.

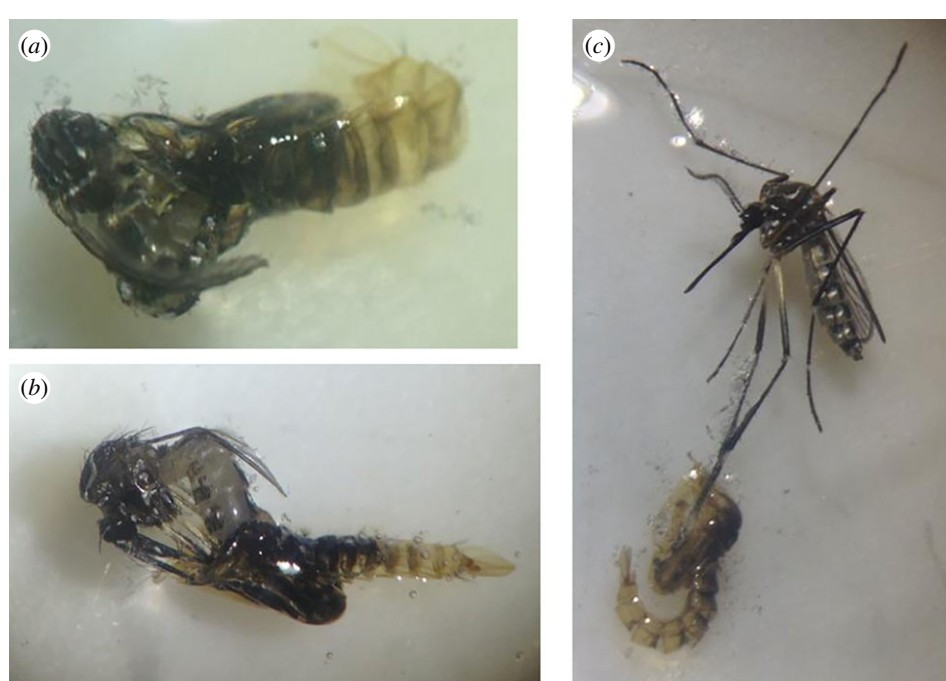

**Figure 5.** Partial transition from pupae to adults of *Ae. aegypti* exposed to IGR diflubenzuron at 0.08 ppm. (*a–c*) The incomplete metamorphosis from the aquatic form to its terrestrial winged form. These photos were taken from the St Catherine population after 5 days of exposure to 0.08 ppm diflubenzuron. (*a,b*) were observed dead, either the entire body (*a*), abdomen (*b*) or leg (*c*) remained fused to the pupal case preventing complete emergence of the adult form.

**Table 2.** Total per cent inhibition of emergence of *Ae. aegypti* larvae exposed to varying concentrations of diflubenzuron. Exposure of *Ae. aegypti* larvae ($N = 60$–110) to varying concentrations (0–0.08 ppm) of diflubenzuron for populations from the eastern parishes (St Catherine, KSA, St Thomas, St Mary, St Ann and Portland) in comparison to the susceptible Rockefeller strain. Data are displayed as total inhibition of emergence for each population (see Material and methods).

| [diflubenzuron] ppm | Rockefeller | St Catherine | KSA | St Thomas | St Mary | St Ann | Portland |
|---|---|---|---|---|---|---|---|
| 0.00013 | 76.67 | 0.00 | 23.20 | 0.00 | 19.89 | 15.79 | 26.67 |
| 0.00064 | 90.00 | 4.00 | 21.69 | 0.00 | 1.50 | 20.05 | 6.67 |
| 0.0032 | 100.00 | 17.00 | 24.14 | 0.00 | 13.32 | 16.77 | 36.67 |
| 0.016 | 100.00 | 23.00 | 16.10 | 0.00 | 28.66 | 21.05 | 100.00 |
| 0.08 | 100.00 | 100.00 | 89.41 | 90.91 | 100.00 | 91.72 | 100.00 |

**Table 3.** Total per cent inhibition of emergence of *Ae. aegypti* larvae exposed to varying concentrations of methoprene. Effect of methoprene (0–0.08 ppm) on the late third instar *Ae. aegypti* larvae ($N = 60$–110) reared from field-collected eggs from the eastern parishes of Jamaica in comparison to the susceptible Rockefeller strain. The data are summarized as total inhibition of emergence per population.

| [methoprene] ppm | Rockefeller | St Catherine | KSA | St Thomas | St Mary | St Ann | Portland |
|---|---|---|---|---|---|---|---|
| 0.00013 | 78.57 | 7.50 | 10.85 | 3.66 | 11.11 | 38.47 | 0.00 |
| 0.00064 | 80.00 | 10.00 | 0.00 | 18.07 | 8.64 | 40.41 | 15.00 |
| 0.0032 | 96.00 | 8.75 | 0.00 | 32.50 | 10.81 | 86.01 | 28.33 |
| 0.016 | 100.00 | 16.25 | 22.79 | 12.20 | 18.06 | 89.99 | 53.00 |
| 0.08 | 100.00 | 16.25 | 92.24 | 90.24 | 45.12 | 100.00 | 36.00 |

should be conducted on a regular basis in countries where vector-borne diseases represent a major burden. Conducting surveillance studies will generate a wealth of information such as baseline resistance levels, and susceptibility trends over time and space, which are necessary for the selection of the best management strategy against *Ae. aegypti* populations. Such recommendation will increase the insecticide/larvicide impact in the field and will also translate into cost-effective operations. Though the concept of insecticide susceptibility was not new to the local authorities in Jamaica, an update of the susceptibility profile for *Ae. aegypti* was needed, as well as the incorporation of an array of alternative products (biolarvicide and IGR) that in time could represent new resources if the traditional larvicides are proven ineffective against the local populations of mosquito vectors.

The susceptibility of the third instar *Ae. aegypti* larvae ($F_0$ generation) from the seven easterly parishes of Jamaica to four [4] larvicides commonly used in vector management [14] was tested. The larvicides—temephos, *Bacillus thuringiensis israelensis* (Bti), diflubenzuron and methoprene—had varying effects, depending on length of exposure, concentration and mode of action of the active ingredients. The toxic effect of the organophosphate temephos and the entomopathogenic (that is, a pathogen that affects insects) bacteria Bti [18] were observed up to 1 h and then at 24 h. The data were interpreted according to the WHO guidelines, meaning larvicide concentrations resulting in greater than 98% mortality in larvae were considered effective [19]. The chitin inhibitor diflubenzuron and juvenile hormone regulator methoprene [20] were observed daily, until total emergence and/or total mortality was observed in the control (0.00 ppm), with data recorded as the inhibition of emergence. The well-known laboratory susceptible strain, Rockefeller, was used as the reference strain to demonstrate the efficacy of the larvicides in susceptible populations. In comparison to the Rockefeller strain, the Jamaican populations tested show resistance to the selected larvicides assessed.

Of all the larvicides tested, the biolarvicide, Bti, had the most toxicological effect at concentrations of 6–8 ppm in all populations tested, regardless of exposure time (figure 3). Mortality greater than 98% was observed at concentrations greater than or equal to 6 ppm. The toxicity to 4 ppm Bti at 24 h was also

effective except for larvae from the St Ann and St Catherine populations, which resulted in mortality between 82 and 87%, respectively. Temephos, a larvicide widely used in many countries [21] in their vector control programmes, including Jamaica [1], appeared effective at a concentration of 0.125 ppm, at an exposure time of 24 h, yielding greater than 97% mortality (figure 2). Temephos was less toxic in the St Ann population yielding mortality greater than $67 \pm 17.54\%$ but less than $88 \pm 2.83\%$ at concentrations of 0.125 and 0.625 ppm, respectively, after 24 h of exposure. Of all the larvicides tested, the growth inhibitors had the least toxic effect on *Ae. aegypti* populations from eastern Jamaica. The concentration that resulted in approximately 90% inhibition of emergence to either diflubenzuron or methoprene was 0.08 ppm, the highest dose tested. One hundred per cent larvae mortality was only observed in three of the six populations exposed to the growth inhibitor diflubenzuron, specifically, St Catherine, St Mary and Portland populations, and only the St Ann population for methoprene (tables 2 and 3, respectively).

According to WHO [16], a population of *Ae. aegypti* is considered susceptible to temephos if 98% mortality is achieved with 0.02 ppm temephos. From the data, all populations from eastern Jamaica were resistant to temephos, with the most resistant *Ae. aegypti* population being St Ann, where only $88 \pm 2.83\%$ mortality was achieved at concentrations of 0.625 ppm. Temephos under the brand Abate® has been used in vector programmes since the 1960s [22] in Jamaica, and many countries have reported resistance to this insecticide, including countries within the Caribbean and Latin America [8,21].

Bti produces a toxic crystal that perforates the digestive system of larvae after consumption [18]. Thus, the toxicity of Bti is directly related to consumption of the toxic crystals. Larvae in this study were not fed while exposed to Bti. However, in field trials, the efficacy of Bti may vary depending on the concentration of the bacteria to food availability and/or pollutants [23] in the environment. Bti is specific for the biological control of mosquitoes and blackflies [24]. Multiple field studies have been conducted using both naturally occurring and proprietor brand Bti. While resistance to Bti is seldom reported in the field [23,25], laboratory resistance over multiple generations greater than $F_{10}$ has been recorded [26,27]. Additionally, commercial Bti is known to have a residual effect [18,23] due to proliferation of the bacteria, but UV rays of solar radiation degrade quickly all the leftover bacterial spores after the product is applied. Recent studies have showed that Bti is not harmful for the environment [28]. With all of this in mind, using the correct dose of Bti is important for successful vector management.

Diflubenzuron clearly demonstrated ability to interfere with metamorphism, thus preventing adult emergence (figures 4 and 5), leading to death as early as 48 h after exposure (figure 4a,c), a phenomenon previously described in *Culex tarsalis* for the growth inhibitor methoprene [29]. These intermediate stages were only observed at 0.08 ppm diflubenzuron, which prevented, on average, 90% adult emergence (table 2). In comparison to other studies, 50% emergence was prevented by diflubenzuron at 0.00036 ppm in *Ae. aegypti* field population from Saudi Arabia [11] and 0.0016 ppm from Argentina [30], while 90% emergence was prevented at 0.0035 ppm in laboratory-reared *Ae. aegypti* [31] and 0.00084 ppm in *Aedes albopictus* from Florida [20]. From the results of this study, diflubenzuron appears ineffective for use in the *Ae. aegypti* management in eastern Jamaica.

Methoprene at 0.08 ppm inhibited 16.25% emergence in the *Ae. aegypti* population from St Catherine and 45.12% emergence in the *Ae. aegypti* population from St Mary and prevented greater than 90% but less than 98% emergence for the *Ae. aegypti* mosquitoes collected in KSA and St Thomas. St Ann *Ae. aegypti* populations were effectively inhibited (table 3) at 0.08 ppm; no variations in morphology were observed in any population subjected to methoprene. Ali *et al*. [20] found that methoprene at 0.008 ppm resulted in 90% inhibition of emergence in *Aedes albopictus*, while Braga *et al*. [32] found that 0.01 ppm caused 90% inhibition of emergence in *Ae. aegypti*. IGR such as diflubenzuron and methoprene are considered to be environmentally friendly as they are biodegradable and considered safe to non-targeted organisms, thus making them an ideal pesticide to use in the field. From the results, however, methoprene may also not be ideal for vector control in the eastern parishes of Jamaica because of its low inhibiting effect to the larvae tested. It should be noted that intolerance to these IGR have previously been reported in field studies in *Aedes* spp. [33].

## 5. Conclusion

Regular insecticide susceptibility surveillance is a necessary tool to incorporate into vector control programmes to detect insecticide resistance at an early stage, adjust insecticide dosage to ensure efficacy and to incorporate new insecticide treatments into existing programmes. In this study, four insecticides were tested, and only Bti at concentrations greater than or equal to 6 ppm produced mortality of 98–

100% in all test *Ae. aegypti* populations. While temephos at concentration 0.625 ppm demonstrated efficacy in five of the six populations, it would be recommended that its use be limited, considering that 0.625 ppm temephos is 31 times greater than the dose recommended by the WHO, and even at that dosage, it appeared ineffective against the St Ann populations. As such, it is the recommendation from this study to incorporate Bti at concentrations of 6–8 ppm into the vector management programme for Jamaica, and that regular susceptibility testing be conducted to detect the development of resistance.

Ethics. The Zika AIRS Project, Jamaica, worked in collaboration with the Ministry of Health and Wellness, Jamaica, to gain the approval from each resident to establish sentinel sites around their homes and to collect samples. Access to resident homes was granted through the period 17 September–7 December 2018. Through this period samples were collected and processed under the project. No special permits were required to establish the sentinel sites nor to use samples collected from the sentinel sites for this study.

Data accessibility. Data available from the Dryad Digital Repository: https://doi.org/10.5061/dryad.g1jwstqn5 [34].

Authors' contributions. S.F.: author, experimental design, data collection and data analysis. J.C.: data collection. S.M., T.C., D.W. and T.H.: data collection. S.H.-J. and S.S.: project design, editor. A.B. and K.A.: Editor. C.T.G.: project design, editor, experimental design, data collection and analysis. All authors gave final approval for publication.

Competing interests. A.B. and K.A. are employed by the USAID as contracting officer representatives (CORs). The CORs reviewed all workplans and subsequent report deliverables under the project; however, they only had an editorial role in the preparation of the manuscript. The technical team led by Abt Associates and the in-country team of the Zika AIRS project (ZAP), with support of the Ministry of Health, designed laboratorial procedures following international and standard protocols for insecticide susceptibility surveillance.

Funding. The activities of Zika AIRS project Jamaica were funded by United States Agency for International Development (USAID), Contract no. AID-GHN-I-00-09-00013; Task Order AID-OAA-TO-14-00035.

Acknowledgements. We acknowledge the great support given by the United States Agency for International Development by funding the Zika AIRS (ZAP) project that implemented entomological monitoring and vector control strategies in Jamaica. Additionally, we thank the Mosquito Control and Research Unit at the University of the West Indies for hosting all the insectary and laboratory procedures required for this study. Finally, we thank our ZAP colleagues who contributed for this study with reviews and key adminstrative support: Jean Margaritis Otto, Abigail Donner, Dasha Migunov and Paula Wood (Project Director). Dr Audrey Lenhart, from Centers for Disease Control and Prevention (CDC), Atlanta, GA, USA, for generously donating the Rockefeller eggs used in this study.

Disclaimer. This manuscript contains results generated from the project. The contents within are the responsibility of the authors and do not necessarily reflect the views of USAID or the United States government. The funders had no role in data collection and analysis, or decision to publish.

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
