## [Reviewer comments · Royal Society Open Science]

Review History

RSOS-192041.R0 (Original submission)

Review form: Reviewer 1 (Victoria Ingham)

Is the manuscript scientifically sound in its present form?

Yes

Are the interpretations and conclusions justified by the results?

Yes

Is the language acceptable?

Yes

Do you have any ethical concerns with this paper?

No

Have you any concerns about statistical analyses in this paper?

No

Recommendation?

Accept with minor revision (please list in comments)

Comments to the Author(s)

An excellently written manuscript on the effects of various larvicides on *Ae. aegyptii* mosquitoes from Jamaica. The manuscript is well written, the data supports the claims and the authors were thorough in their tests. The statistical methodologies are good.

Just very small notes:

I can't see the y-axis lines in any of the figures

Rockerfeller is spelt wrong on line 220

The authors specify N as number of mosquitoes but don't state number of replicates

Figure 3 should just have a small note on what the arrows show in the legend, doesn't need to be too descriptive as it is mentioned in the text

The authors should remove the numbers in brackets, unless this is a journal requirement as it is confusing with the references being in the same style

Review form: Reviewer 2 (Silvie Huijben)

Is the manuscript scientifically sound in its present form?

No

Are the interpretations and conclusions justified by the results?

Yes

Is the language acceptable?

Yes

Do you have any ethical concerns with this paper?

No

Have you any concerns about statistical analyses in this paper?

Yes

Recommendation?

Major revision is needed (please make suggestions in comments)

Comments to the Author(s)

This is a thorough study on the insecticide susceptibility levels of various larvicides in Jamaica. The study itself was well-executed with different dosages and incorporating a lab reference strain (but see major comments below). While the results are not ground breaking, they contribute to an important body of literature of insecticide susceptibility regionally and globally. However, I do have some major comments and concerns regarding the presentation of the methods and the results, and would not be able to recommend this manuscript for publication if these are not addressed. In addition, I have many minor comments to improve clarity of the manuscript.

Major comments:

- Next to providing the percentages (in the supplementary material), also provide the raw data so that the reader can have access to the number of larvae tested as well. This is particularly important because ranges of larvae being used are given. In addition, it provides credibility to the data.

- Related to that, I would advise to present the error margins of the data in the form of standard error of the mean, rather than standard deviation. This is to account for differences in sample numbers between different sites. As the aim of the study is to test for differences between means, the standard error is a better metric.
- In the conflict of interest statement, it states that authors A, B and K. A only had an editorial role, but under “author contributions” it also states that these authors contributed to project design. Please comment to whether this leads to a conflict of interest with both these authors being employed by USAID. Either way, both these sections of the paper should agree with each other and not contradict.
- It is great that a lab reference strain is used as a positive control to ascertain that any deviation from 98-100% mortality is due to the mosquito strain tested and no outside factors. However, there is no information given about the procedure of this strain in the methods or even mentioned. The specifics of this part of the study should be clearly defined. Most importantly, are these experiments performed at the same time, in the same lab with the same dilutions by the same people?
- It would be useful to have a map of the study sites to compare the results with geographical locations.
- There is no analysis on the difference between the reference strain and the field strains. This should be the first analysis, which may then be followed by testing for differences between the sites.
- Line 369: The conclusion states thirteen populations, but all previous mentions of the number of populations/parishes only states seven (including the tables and graphs).
- There’s inconsistency in the various tables with how many decimals are given and when letters are given as comparison between sites. Also, no statistics and st.dev or standard errors are given in table 2. This may be because total emergence per treatment concentration was calculated, but this should be emergence per replicate as with the other insecticides.
- Figure 1: “means with different letters are significantly different at $p < 0.005$ ” (line 202). Is this a typo and should it be 0.05 (according to methods)? Same on line 206.

Minor comments:

- Line 69: Remove “(7)”. This is confusing to the reader because it looks like a citation. (Same for lines 85, 106, 190, 287, and 310)
- Line 73: Replace “it” with “effectiveness”
- Line 76-77: Some more details on this would be useful to the reader! Also, replace “hematophagous” with “blood-feeding.”
- Line 79: Is Bti considered a chemical? Would this be included in “chemical resistance?”
- Line 84-89: This should be moved to methods.
- Line 85: Insert “of the seven” to be more clear that one of the parishes did not receive this treatment.
- Line 92: Replace “insecticides” with “larvicides”
- Line 98-104: Move this to lower in the methods where it describes the exposures.
- Line 105: It would be useful to start with a ‘study location’ paragraph with information on geographical location, climate, typical mosquito season (in relation to when collections were done), abundance, arbovirus transmission etc. Brief but useful.
- Line 114-115: How long were the egg papers allowed to dry? How long were eggs stored before they were hatched?
- Line 124: Spell out hours; Replace “light: dark” with “light:dark” (remove extra space).
- Lines 127-128: Add more detail: Were eggs or larvae collected? Where? What stage? Was there a different protocol than the other sites? Merge with paragraph ‘Oviposition’.
- Line 132: Clarify that the two insect growth regulators were not applied together.
- Line 137: Did the insecticide treated cups also contain 0.4% ethanol?
- Line 138-139, 149: Why did you first observe each assay for one hour? Was it for quality control? Is this standard protocol?
- Line 143: “prepare” (Prepare misspelled)
- Line 146: Provide details on which concentrations were tested, rather than a range.
- Line 147: Why are more larvae used for Bti larvicide testing than the other test groups?

- Line 166: Reference 14 is cited only here. Consider citing it along with reference 13.
- Lines 170-172: Revise. These two sentences are confusing.
- Line 176-284: Unclear why similar data is sometimes presented in table and sometimes in figure without obvious logic. Could be done more consistently.
- Lines 178, 229, 270: Tables 1, 2, and 3:
 - o Be consistent with all three tables with the placement of the populations in either the first column or the first row.
 - o Align the decimal points within each column (especially for Table 1)
 - o Vertically center all text within each cell.
 - o Widen cells so that each cell can display the full number in one line (and not two). See Table 3, column 4.
- Line 183: Remove the comma after "Portland" (Same for line 216)
- Lines 194, 211: Figure 1, 2A, and 2B:
 - o Consider added line hatching on the different Temephos/Bti concentrations to help distinguish them apart from each other more clearly for people who print in black and white and/or for people who are color blind.
 - o Use black font for text here to match all other text and to read more easily.
- Line 198: Write the order of the parishes to match that of within the rest of the manuscript (ex: line 183, lines 215-216, and the tables and figures).
- Line 221: Within one hour of exposure
- Line 221-222: "No significant differences were observed at these concentrations" for both the one hour and 24 hour exposure?
- Line 237: Figure 3: This figure needs more details to be more useful. Parts of the text on lines 247-251 could be moved to the legend. It is not clear what the red arrows are precisely referring to. For instance, the normal pupa in figure 3d appears to also have anal gills? Or are these different structures? It is great to see these pictures but they could be more clarified. Figure 3c is very blurry.
- Line 262: Insert "and" and remove the comma after "c" so that it reads "Figures 4a, b, and c show..."
- Lines 266-269: Pictures 4a and 4b need details like are given for 4c.
- Line 291: Use of "entomopathogenic" - will your audience know what this means?
- Line 307: Be more specific with which mosquitoes, because you only tested *Aedes aegypti* and not all mosquitoes within eastern Jamaica.
- Line 325: Spell out environments
- Line 328: What is Bti known to have a residual effect on? The ecosystem? All organisms?
- Lines 331-337: Repeat paragraph from lines 313-319
- Line 353: Revise; "no advancement in morphology..."
- Line 368: Replace "*Ae. aegypti*" with "*A. aegypti*" (or vice versa) to match all other occurrences in the manuscript.
- Line 371: Introduce the recommended/diagnostic dose earlier in the article. It was only given in the conclusion and that would have been useful information to interpret the figures shown before. Indicating it visually in the figures would also be great.
- Line 379: Move "(ZAP)" to after "ZIKA AIRS Project". This will then match line 66.
- Line 385: Replace "for" with "to"
- Line 440: Repeat of year "2005"

Decision letter (RSOS-192041.R0)

03-Feb-2020

Dear Dr Francis

On behalf of the Editors, I am pleased to inform you that your Manuscript RSOS-192041 entitled

"Comparative toxicity of larvicides and growth inhibitors on *Aedes aegypti* from select areas in Jamaica" has been accepted for publication in Royal Society Open Science subject to minor revision in accordance with the referee suggestions. Please find the referees' comments at the end of this email.

The reviewers and handling editors have recommended publication, but also suggest some minor revisions to your manuscript. Therefore, I invite you to respond to the comments and revise your manuscript.

- Ethics statement

- Data accessibility

If you wish to submit your supporting data or code to Dryad (<http://datadryad.org/>), or modify your current submission to dryad, please use the following link:
<http://datadryad.org/submit?journalID=RSOS&manu=RSOS-192041>

- Competing interests

- Authors' contributions

- Acknowledgements

- Funding statement

Because the schedule for publication is very tight, it is a condition of publication that you submit the revised version of your manuscript before 12-Feb-2020. Please note that the revision deadline will expire at 00.00am on this date. If you do not think you will be able to meet this date please let me know immediately.

If your manuscript is newly submitted and subsequently accepted for publication, you will be asked to pay the article processing charge, unless you request a waiver and this is approved by Royal Society Publishing. You can find out more about the charges at <https://royalsocietypublishing.org/rsos/charges>. Should you have any queries, please contact openscience@royalsociety.org.

Kind regards,

Anita Kristiansen
Editorial Coordinator

on behalf of Dr Krijn Paaijmans (Associate Editor) and Kevin Padian (Subject Editor)
openscience@royalsociety.org

Associate Editor Comments to Author (Dr Krijn Paaijmans):

Dear author,

After reviewing the two reports shared with me, I decided to accept the manuscript with minor revisions. The major revisions suggested by reviewer #2 seem relatively easy to address. The recommendations regarding the raw data, standard errors, suggestions for analysis (on the difference between the reference strain and the field strains), etc. need to be followed up on.

Reviewer comments to Author:

Reviewer: 1

Comments to the Author(s)

An excellently written manuscript on the effects of various larvicides on *Ae. aegyptii* mosquitoes from Jamaica. The manuscript is well written, the data supports the claims and the authors were thorough in their tests. The statistical methodologies are good.

Just very small notes:

I can't see the y-axis lines in any of the figures

Rockerfeller is spelt wrong on line 220

The authors specify N as number of mosquitoes but don't state number of replicates

Figure 3 should just have a small note on what the arrows show in the legend, doesn't need to be too descriptive as it is mentioned in the text

The authors should remove the numbers in brackets, unless this is a journal requirement as it is confusing with the references being in the same style

Reviewer: 2

Comments to the Author(s)

This is a thorough study on the insecticide susceptibility levels of various larvicides in Jamaica. The study itself was well-executed with different dosages and incorporating a lab reference strain (but see major comments below). While the results are not ground breaking, they contribute to an important body of literature of insecticide susceptibility regionally and globally. However, I do have some major comments and concerns regarding the presentation of the methods and the results, and would not be able to recommend this manuscript for publication if these are not addressed. In addition, I have many minor comments to improve clarity of the manuscript.

Major comments:

- Next to providing the percentages (in the supplementary material), also provide the raw data so that the reader can have access to the number of larvae tested as well. This is particularly important because ranges of larvae being used are given. In addition, it provides credibility to the data.
- Related to that, I would advise to present the error margins of the data in the form of standard error of the mean, rather than standard deviation. This is to account for differences in sample numbers between different sites. As the aim of the study is to test for differences between means, the standard error is a better metric.
- In the conflict of interest statement, it states that authors A. B and K. A only had an editorial role, but under "author contributions" it also states that these authors contributed to project design. Please comment to whether this leads to a conflict of interest with both these authors being employed by USAID. Either way, both these sections of the paper should agree with each other and not contradict.
- It is great that a lab reference strain is used as a positive control to ascertain that any deviation from 98-100% mortality is due to the mosquito strain tested and no outside factors. However, there is no information given about the procedure of this strain in the methods or even mentioned. The specifics of this part of the study should be clearly defined. Most importantly, are these experiments performed at the same time, in the same lab with the same dilutions by the same people?
- It would be useful to have a map of the study sites to compare the results with geographical locations.
- There is no analysis on the difference between the reference strain and the field strains. This should be the first analysis, which may then be followed by testing for differences between the sites.
- Line 369: The conclusion states thirteen populations, but all previous mentions of the number of populations/parishes only states seven (including the tables and graphs).
- There's inconsistency in the various tables with how many decimals are given and when letters are given as comparison between sites. Also, no statistics and st.dev or standard errors are given in table 2. This may be because total emergence per treatment concentration was calculated, but this should be emergence per replicate as with the other insecticides.
- Figure 1: "means with different letters are significantly different at $p < 0.005$ " (line 202). Is this a typo and should it be 0.05 (according to methods)? Same on line 206.

Minor comments:

- Line 69: Remove "(7)". This is confusing to the reader because it looks like a citation. (Same for lines 85, 106, 190, 287, and 310)
- Line 73: Replace "it" with "effectiveness"
- Line 76-77: Some more details on this would be useful to the reader! Also, replace "hematophagous" with "blood-feeding."
- Line 79: Is Bti considered a chemical? Would this be included in "chemical resistance?"
- Line 84-89: This should be moved to methods.
- Line 85: Insert "of the seven" to be more clear that one of the parishes did not receive this treatment.

- Line 92: Replace “insecticides” with “larvicides”
- Line 98-104: Move this to lower in the methods where it describes the exposures.
- Line 105: It would be useful to start with a ‘study location’ paragraph with information on geographical location, climate, typical mosquito season (in relation to when collections were done), abundance, arbovirus transmission etc. Brief but useful.
- Line 114-115: How long were the egg papers allowed to dry? How long were eggs stored before they were hatched?
- Line 124: Spell out hours; Replace “light: dark” with “light:dark” (remove extra space).
- Lines 127-128: Add more detail: Were eggs or larvae collected? Where? What stage? Was there a different protocol than the other sites? Merge with paragraph ‘Oviposition’.
- Line 132: Clarify that the two insect growth regulators were not applied together.
- Line 137: Did the insecticide treated cups also contain 0.4% ethanol?
- Line 138-139, 149: Why did you first observe each assay for one hour? Was it for quality control? Is this standard protocol?
- Line 143: “prepare” (Prepare misspelled)
- Line 146: Provide details on which concentrations were tested, rather than a range.
- Line 147: Why are more larvae used for Bti larvicide testing than the other test groups?
- Line 166: Reference 14 is cited only here. Consider citing it along with reference 13.
- Lines 170-172: Revise. These two sentences are confusing.
- Line 176-284: Unclear why similar data is sometimes presented in table and sometimes in figure without obvious logic. Could be done more consistently.
- Lines 178, 229, 270: Tables 1, 2, and 3:
 - o Be consistent with all three tables with the placement of the populations in either the first column or the first row.
 - o Align the decimal points within each column (especially for Table 1)
 - o Vertically center all text within each cell.
 - o Widen cells so that each cell can display the full number in one line (and not two). See Table 3, column 4.
- Line 183: Remove the comma after “Portland” (Same for line 216)
- Lines 194, 211: Figure 1, 2A, and 2B:
 - o Consider added line hatching on the different Temephos/Bti concentrations to help distinguish them apart from each other more clearly for people who print in black and white and/or for people who are color blind.
 - o Use black font for text here to match all other text and to read more easily.
- Line 198: Write the order of the parishes to match that of within the rest of the manuscript (ex: line 183, lines 215-216, and the tables and figures).
- Line 221: Within one hour of exposure
- Line 221-222: “No significant differences were observed at these concentrations” for both the one hour and 24 hour exposure?
- Line 237: Figure 3: This figure needs more details to be more useful. Parts of the text on lines 247-251 could be moved to the legend. It is not clear what the red arrows are precisely referring to. For instance, the normal pupa in figure 3d appears to also have anal gills? Or are these different structures? It is great to see these pictures but they could be more clarified. Figure 3c is very blurry.
- Line 262: Insert “and” and remove the comma after “c” so that it reads “Figures 4a, b, and c show...”
- Lines 266-269: Pictures 4a and 4b need details like are given for 4c.
- Line 291: Use of “entomopathogenic” - will your audience know what this means?
- Line 307: Be more specific with which mosquitoes, because you only tested *Aedes aegypti* and not all mosquitoes within eastern Jamaica.
- Line 325: Spell out environments
- Line 328: What is Bti known to have a residual effect on? The ecosystem? All organisms?
- Lines 331-337: Repeat paragraph from lines 313-319
- Line 353: Revise; “no advancement in morphology...”
- Line 368: Replace “*Ae. aegypti*” with “*A. aegypti*” (or vice versa) to match all other occurrences in the manuscript.

- Line 371: Introduce the recommended/ diagnostic dose earlier in the article. It was only given in the conclusion and that would have been useful information to interpret the figures shown before. Indicating it visually in the figures would also be great.
- Line 379: Move "(ZAP)" to after "ZIKA AIRS Project". This will then match line 66.
- Line 385: Replace "for" with "to"
- Line 440: Repeat of year "2005"

Author's Response to Decision Letter for (RSOS-192041.R0)

See Appendix A.

Decision letter (RSOS-192041.R1)

24-Feb-2020

Dear Dr Francis,

It is a pleasure to accept your manuscript entitled "Comparative toxicity of larvicides and growth inhibitors on *Aedes aegypti* from select areas in Jamaica" in its current form for publication in Royal Society Open Science. The comments of the reviewer(s) who reviewed your manuscript are included at the foot of this letter.

Kind regards,
Andrew Dunn
Royal Society Open Science Editorial Office

on behalf of Dr Krijn Paaijmans (Associate Editor) and Kevin Padian (Subject Editor)
openscience@royalsociety.org

Appendix A

19th February 2020

Dr. Anita Kristiansen
Editorial Coordinator
Royal Society Open Science

Dear Dr. Anita Kristiansen,

Re: Manuscript ID RSOS-192041

Please find attached our revised manuscript entitled “Comparative toxicity of larvicides and growth inhibitors on *Aedes aegypti* from select areas in Jamaica” (Manuscript ID RSOS-192041) which we have revised in light of the comments made by the reviewers.

We are grateful for the time spent by the reviewers in assessing our manuscript and wish to acknowledge the very helpful comments made. Each comment/suggestion has been addressed below. All changes made to the original submission has been highlighted by “track changes” in Microsoft word.

My collaborators and I hope that we have addressed all issues satisfactorily and look forward to hearing back from you.

Yours faithfully,

.....
Sheena Francis

Our comments in response to that of the reviewer is written in deep blue below.

Comments addressed for revised manuscript:

Reviewer #1

- I can't see the y-axis lines in any of the figures

Thank you for pointing this out, the text has now been amended

Rockerfeller is spelt wrong on line 220

Thank you for pointing this out, the text has now been amended

- Figure 3 should just have a small note on what the arrows show in the legend, doesn't need to be too descriptive as it is mentioned in the text

This was updated as requested, please refer to line 265 – 267.

- The authors should remove the numbers in brackets, unless this is a journal requirement as it is confusing with the references being in the same style

Thank you for pointing this out, the text has now been amended

Reviewer #2

- Next to providing the percentages (in the supplementary material), also provide the raw data so that the reader can have access to the number of larvae tested as well. This is particularly important because ranges of larvae being used are given. In addition, it provides credibility to the data.
 - Related to that, I would advise to present the error margins of the data in the form of standard error of the mean, rather than standard deviation. This is to account for differences in sample numbers between different sites. As the aim of the study is to test for differences between means, the standard error is a better metric.

Thank you for pointing this out, the document has now been amended

- In the conflict of interest statement, it states that authors A. B and K. A only had an editorial role, but under “author contributions” it also states that these authors contributed to project design. Please comment to whether this leads to a conflict of interest with both these authors being employed by USAID. Either way, both these sections of the paper should agree with each other and not contradict.

Thanks you for pointing this out, the statement will be updated as stated below in the corresponding section of the submission.

“Authors A.B and K.A are employed to the USAID as Contracting officer representatives (CORs). The CORs reviewed all workplans and subsequent report deliverables under the

project, however they only had an editorial role in the preparation of the manuscript. The technical team led by Abt Associates and the in country team of the Zika AIRS project (ZAP), with support of the Ministry of Health, designed laboratorial procedures following international and standard protocols for insecticide susceptibility surveillance.”

- It is great that a lab reference strain is used as a positive control to ascertain that any deviation from 98-100% mortality is due to the mosquito strain tested and no outside factors. However, there is no information given about the procedure of this strain in the methods or even mentioned. The specifics of this part of the study should be clearly defined. Most importantly, are these experiments performed at the same time, in the same lab with the same dilutions by the same people?

We have amended the methods section, please see lines 109 – 113, the authors have included the origin of the reference strain (donated by CDC). The procedures follow the WHO and CDC protocols. The reference strain (Rockefeller) was kept isolated from other populations the entire time.

In Regards to the procedure on how the experiments were conducted per type of product. All replicates of larvicides tested were evaluated at the same time. Bioassays were scheduled as the number of individuals were available – in order to use the number of larvae instructed by WHO protocols. All procedures were conducted in the same lab –the Mosquito Control and Research Unit (MCRU) located in the campus of the University of the West Indies (UWI, Mona Campus). The MCRU was consolidated with the USAID funding and the institutional agreement of the UWI and the Ministry of Health of Jamaica. The Laboratory procedures and larvicide testing was conducted by Dr. Sheena Francis. Dr. Francis led a team of laboratory technicians in charge of the mosquito colony. All dilutions were prepared by Dr. Francis, all mortality records were supervised by Dr. Francis.

- There is no analysis on the difference between the reference strain and the field strains. This should be the first analysis, which may then be followed by testing for differences between the sites.

We thank the reviewer for pointing this omission out.

We conducted an analysis of variance, which included the reference strain and the Jamaican populations for all bioassays per concentration. After which we conducted *post hoc* analysis and chose to report significant findings between the test populations. Particularly in light that all the calculated percent mortalities of the Jamaican populations were statistically different from the reference strain. Also, larvae populations from Jamaica specifically from St Andrew, had been previously reported as being resistant. This we quoted in the manuscript. Instead we chose visual displays in the form of charts and in tables to show the marked difference.

Again, we thank the reviewer for highlighting the omission. The manuscript has now been modified and we hope that we have made the reader aware that the comparison between the reference strain and the tested Jamaican population was made.

- It would be useful to have a map of the study sites to compare the results with geographical locations.

We agree with the reviewer and have made the addition of a map. A map of Jamaica highlighting the locations of mosquito sampling has been added as figure 1.

- Line 369: The conclusion states thirteen populations, but all previous mentions of the number of populations/parishes only states seven (including the tables and graphs).

We thank the reviewer for pointing this error out, the manuscript has been amended in line 406.

- There's inconsistency in the various tables with how many decimals are given and when letters are given as comparison between sites.

Thank you for pointing this out, the tables have now been amended

- Also, no statistics and st.dev or standard errors are given in table 2. This may be because total emergence per treatment concentration was calculated, but this should be emergence per replicate as with the other insecticides.

For either of the growth regulator, Methoprene or Diflubenzuron, the replicate results for each concentration was pooled to give a total mortality. We have rearranged the how sentences highlighting this issue was written and we now hope that the modifications throughout the document makes this point clearer.

- Figure 1: "means with different letters are significantly different at $p < 0.005$ " (line 202). Is this a typo and should it be 0.05 (according to methods)? Same on line 206.

We would like to thank the reviewer for picking this up, however, the line is correct. The figure states the calculated statistical significance, which was greater than 0.05.

- Line 69: Remove "(7)". This is confusing to the reader because it looks like a citation. (Same for lines 85, 106, 190, 287, and 310)

We agree with the reviewer, these numbers in parentheses were removed from the respective lines.

- Line 73: Replace "it" with "effectiveness"

The manuscript has now been updated with effectiveness instead of it, line 71.

- Line 76-77: Some more details on this would be useful to the reader! Also, replace "hematophagous" with "blood-feeding."

Respectfully, we the authors would rather keep the original word selection (hematophagous) as we believe that it fits well with the audience of this scientific journal.

- Line 79: Is Bti considered a chemical? Would this be included in “chemical resistance?”

Bti is considered a bio-larvicide. The text of the manuscript has now been modified to ensure this concept is coherent with the narrative.

- Line 84-89: This should be moved to methods.

The final paragraph of the introduction was modified to correct this issue. The methodology section was also edited to incorporate the content relevant to field and lab procedures. Thank you for pointing out this.

- Line 98-104: Move this to lower in the methods where it describes the exposures.

The authors considered that the kit of larvicides is appropriately described under the “Materials” section – as in this paragraph we are establishing that such elements were purchased from authorized institutions. Later in the methods the text focused on concentrations of larvicides only.

- Line 105: It would be useful to start with a ‘study location’ paragraph with information on geographical location, climate, typical mosquito season (in relation to when collections were done), abundance, arbovirus transmission etc. Brief but useful.

This has been modified according to the reviewer’s suggestion. An entire new paragraph was added from line 89.

- Line 114-115: How long were the egg papers allowed to dry? How long were eggs stored before they were hatched?

The eggs were initially allowed to air dry for a few days. Once dried, some of the eggs were stored for a week – when needed- inside properly labeled Ziploc bags. The vast majority of the biological material was activated right after it was brought from the field given the high demand for live larvae during larvicide bioassays. We have updated the manuscript for clarity, thank you.

- Line 124: Spell out hours; Replace “light: dark” with “light:dark” (remove extra space).

This edit has been made, again we thank the reviewer.

- Lines 127-128: Add more detail: Were eggs or larvae collected? Where? What stage? Was there a different protocol than the other sites? Merge with paragraph ‘Oviposition’.

The wild eggs collected are mentioned in the part describing the oviposition from line 114. Additionally, the Ministry of Health and Wellness of Jamaica donated wild larvae from one Parish (Portland). This information was originally included in the text as “The Larvae from

Portland were field caught and donated by the Ministry of Health and Wellness, Jamaica to the project. The larvae were reared as described above. The insectary facilities employed by this study correspond to the recently inaugurated Mosquito Control and Research Unit at the University of West Indies, Mona campus, in Kingston, Jamaica. The authors believe such topics have been clearly included in the body of the manuscript.

- Line 132: Clarify that the two insect growth regulators were not applied together.

The authors have stated that the laboratorial procedures followed the protocols of the World Health Organization (WHO), the instructions of which state that each product should have a different bioassay. We have described with photo, tables and in the discussion the effect of each growth regulator separately. On reviewing the manuscript, we respectfully believe that it states that the assays were conducted separately. We have however made some amendments and do hope that the changes make the point clearer.

- Line 137: Did the insecticide treated cups also contain 0.4% ethanol?

The kits for larvicide evaluation are complete kits with full doses for the products (larvicides) and control. These kits were standard WHO kits which came with ingredients and dilution instructions. According to the kit details on product dilution, the larvicides would have been diluted in ethanol.

- Line 138-139, 149: Why did you first observe each assay for one hour? Was it for quality control? Is this standard protocol?

Considering that Bti was to be observed at 1 hour, all experiments were treated similarly.

- Line 143: "prepare" (Prepare misspelled)

Thanks for pointing this out, the manuscript has now been updated

- Line 146: Provide details on which concentrations were tested, rather than a range.

The text has now been modified to provide the individual concentrations of Bti (0, 2ppm, 4ppm, 6ppm, 8ppm). Please see lines 166 to 167.

- Line 147: Why are more larvae used for Bti larvicide testing than the other test groups?

The average number of larvae exposed to the different products was over 75, these number were dependent on the amount of *Ae. aegypti* larvae available for each bioassays. The larvae from Portland was gifted to us from the Ministry of health and wellness, Jamaica, the assays reported used 20 larvae or more from Portland owing to availability of the larvae. We thank the reviewer for point this out, the manuscript has now been updated

- Line 166: Reference 14 is cited only here. Consider citing it along with reference 13. Thanks for pointing this out, the manuscript has now been updated

- Line 291: Use of "entomopathogenic" - will your audience know what this means?

We have added the meaning for this word. See lines 333.

- Line 325: Spell out environments

Thanks for pointing this out, the manuscript has now been updated

- Line 328: What is Bti known to have a residual effect on? The ecosystem? All organisms?

An explanation has now been added, and references can be found in lines 369 - 376.

- Line 176-284: Unclear why similar data is sometimes presented in table and sometimes in figure without obvious logic. Could be done more consistently.
 - Lines 178, 229, 270: Tables 1, 2, and 3:
 - o Be consistent with all three tables with the placement of the populations in either the first column or the first row.
 - o Align the decimal points within each column (especially for Table 1)
 - o Vertically center all text within each cell.
 - o Widen cells so that each cell can display the full number in one line (and not two). See Table 3, column 4.
 - Lines 194, 211: Figure 1, 2A, and 2B:
 - o Consider added line hatching on the different Temephos/Bti concentrations to help distinguish them apart from each other more clearly for people who print in black and white and/or for people who are color blind.
 - o Use black font for text here to match all other text and to read more easily.
 - Line 198: Write the order of the parishes to match that of within the rest of the manuscript (ex: line 183, lines 215-216, and the tables and figures).
 - Line 221: Within one hour of exposure

Thanks for pointing this out, modifications have now been made throughout the manuscript to address these points regarding the tables and figures, also the order in how the population names were written.

- Line 221-222: “No significant differences were observed at these concentrations” for both the one hour and 24 hour exposure?

The organisms were dead at 1 hour and there was no significant difference at that time. The organisms remained dead until 24 hours. The line has been modified to reflect this point, thank you.

Line 237: Figure 3: This figure needs more details to be more useful. Parts of the text on lines 247-251 could be moved to the legend. It is not clear what the red arrows are precisely referring to. For instance, the normal pupa in figure 3d appears to also have anal gills? Or are these different structures? It is great to see these pictures but they could be more clarified. Figure 3c is very blurry.

- Line 262: Insert “and” and remove the comma after “c” so that it reads “Figures 4a, b, and c show...”
- Lines 266-269: Pictures 4a and 4b need details like are given for 4c.

Thank you for your comment, we agree with the reviewer and have modified the manuscript to address this point.

In regards to Figure 3c now 4c we agree with the reviewer that the body of the pupae is blurry, however the areas that we are highlighting are clear, we do not have another picture to replace it with. If the picture is too blurry for print we will remove it.

- Line 353: Revise; “no advancement in morphology...”

The text has now been changed from “no deviations in morphology” to “no variations in morphology” (Line 391)

- Line 368: Replace “*Ae. aegypti*” with “*A. aegypti*” (or vice versa) to match all other occurrences in the manuscript.

All occurrences have now been updated to include the abbreviation of the species uniformly across the entire text. Thank you.

- Line 371: Introduce the recommended/diagnostic dose earlier in the article. It was only given in the conclusion and that would have been useful information to interpret the figures shown before. Indicating it visually in the figures would also be great.

Thanks for this comment, amendments have been made through the document and new paragraph was added, in order to start the discussion regarding regular surveillance of larvicides/insecticides against *Ae. aegypti*.

- Line 379: Move “(ZAP)” to after “ZIKA AIRS Project”. This will then match line 66.

Thanks for pointing this out, the manuscript has now been updated

- Line 385: Replace “for” with “to”

Thanks for pointing this out, the manuscript has now been updated

- Line 440: Repeat of year “2005” done

Thanks for pointing this out, the manuscript has now been updated

We do hope that the manuscript reads clearer.

Sincerely

.....
Sheena Francis